# Visible-Light-Activated Carbon Dot Photocatalyst for ROS-Mediated Inhibition of Algae Growth

**DOI:** 10.3390/ijms241713509

**Published:** 2023-08-31

**Authors:** Jun Song, Zhibin Xu, Hao Li, Yu Chen, Jiaqing Guo

**Affiliations:** State Key Laboratory of Radio Frequency Heterogeneous Integration (Shenzhen University), College of Physics and Optoelectronic Engineering, Key Laboratory of Optoelectronic Devices and Systems of Ministry of Education and Guangdong Province, Shenzhen University, Shenzhen 518060, China; songjun@szu.edu.cn (J.S.); misakasagiri@gmail.com (Z.X.); gjq@szu.edu.cn (J.G.)

**Keywords:** carbon dot, reactive oxygen species, harmful algae bloom, FLIM

## Abstract

The growing occurrence of detrimental algal blooms resulting from industrial and agricultural activities emphasizes the urgency of implementing efficient removal strategies. In this study, we have successfully synthesized stable and biocompatible carbon dots (R-CDs) capable of generating reactive oxygen species (ROS) upon exposure to natural light irradiation. Phaeocystis globosa Scherffel (PGS) was selected as a representative model for conducting anti-algal experiments. Remarkably, in the presence of R-CDs, the complete eradication of harmful algae within a simulated light exposure period of 27 h was achieved. Furthermore, fluorescence lifetime imaging microscopy (FLIM) was first employed to study the physiological processes involved in the oxidative stress induced by PGS when subjected to ROS attack. The findings of this study demonstrate the potential of R-CDs as a highly promising anti-algal agent. This elucidation of the mechanism contributes to a comprehensive understanding of the efficacy and effectiveness of such agents in combating algal growth, further inspiring the development of other anti-algal agents.

## 1. Introduction

Algal blooms caused by eutrophication and climate change have become a critical issue and have led to water quality deterioration and the death of aquatic organisms [1,2,3,4]. HABs (harmful algal blooms) can severely affect the biodiversity of bodies of water, fisheries, tourism, and public safety. Therefore, finding effective ways to address HABs is essential [5,6,7,8].

The methods used to control HABs can be roughly divided into physical, chemical, and biological [9]. Physical methods, such as algae fishing, ultrasonic algal killing, UV irradiation, and pumping mixed water layers, have the advantage of quick results [10,11,12,13], but their labor costs are extremely high; therefore, they can only be used in small bodies of water. Chemical methods based on chemical algaecides and flocculants are cost-effective but can also cause secondary environmental pollution [14,15,16,17]. Biological control is an environmentally friendly approach that reduces the dominant position of harmful algae in the aquatic environment through predation, parasitism, and competition, thereby achieving a new ecological balance [18,19,20]. However, in practical applications, the long effective time and the risk of species invasion limit its effectiveness [21]. Therefore, methods capable of overcoming these drawbacks possess significant potential for broad application.

With the discovery that TiO_2_ decomposes water under light and oxidative stress in algae, the use of metal nanoenzymes has received increased attention. The use of R-CDs combined with FLIM can be a new research method. Photocatalytically resisting algae has gained widespread attention [22,23,24]. Currently, nanoparticles such as TiO_2_, silver, and copper oxide have demonstrated remarkable efficacy in the field of algae resistance [25,26,27,28]. However, the release of metal ions into aquatic environments poses a pollution threat [29,30]. Therefore, the development of a non-metallic photocatalytic nanoenzyme that can stably resist algae is of considerable significance [31,32].

CDs, a class of non-metallic nanomaterials, have attracted significant attention due to their unique properties, including tunable fluorescence, excellent stability, high photoactivity, and low toxicity [33]. Their applications span from photocatalysis and sensitive sensing to advanced biological imaging [34,35,36,37,38,39,40]. As photocatalysts, carbon dots, including graphene quantum dots (GQDs), carbon quantum dots (CQDs), carbon nanodots (CNDs), and carbonized polymer dots (CPDs), have attracted much attention in the field of photo fungicides due to their ability to generate ROS under light, good biocompatibility, good solubility in water, and light bleaching resistance [41,42,43]. Reports of using photodynamic therapy to eliminate harmful units such as bacterial cells and cancer cells are often mentioned [44,45,46,47]. However, the utilization of CDs as non-metallic photocatalysts for anti-algal applications has rarely been reported. On the other hand, research and development on the oxidative stress process of algae are, in relative terms, lagging behind.

In this work (Figure 1), novel R-CDs capable of producing ROS under natural light were purposefully designed and synthesized through a simple hydrothermal method. The excellent catalytic activity and environmental stability of the R-CDs were demonstrated across a range of conditions, encompassing diverse temperatures, pH values, salt concentrations, and ionic solutions. Subsequently, we developed a model to assess the anti-algal activity of R-CDs using the representative HAB-causing algal species PGS as our target organism. We assessed the inhibitory effect of R-CDs on algal proliferation and conducted an in-depth investigation into the mechanism underlying the anti-algal activity of R-CDs by using confocal laser scanning microscopy and fluorescence lifetime imaging microscopy. Our discovery of the ROS-based anti-algal mechanism will significantly contribute to the development of a wider range of biocompatible anti-algal agents, providing effective solutions for addressing the HAB issue.

## 2. Results and Discussion

TEM was used to analyze the morphology and size distribution of the R-CDs. As shown in Figure 1b, most of the synthesized R-CDs were spherical with a uniform dispersion. The particle size distribution of the R-CDs is shown in Figure 1b, which shows the statistics for 100 particles. The particle size ranged from 1.75 to 5.25 nm with an average size of 3.37 ± 0.833 nm. To investigate the structural composition of the R-CDs, FT-IR analysis was conducted to determine the types of functional groups present on their surfaces. As is shown in Figure 1c, the obtained results indicate that the R-CDs exhibit an absorption peak at 3454.99 cm^−1^ [48], which corresponds to the tensile vibration of O-H functional groups. Additionally, the presence of an absorption peak at 1635.89 cm^−1^ indicates the presence of C=C functional groups [49]. These findings provide insights into the chemical structure of the R-CDs, which can be used to further understand their properties and potential applications. The chemical structure of the R-CDs is the main factor affecting their optical properties. XPS was used to determine their chemical composition and functional groups. As shown in the full XPS spectrum of the R-CDs (Figure 1d), five main characteristic peaks were observed at binding energies of 169.27, 284.95, 401.32, 532.21 and 685.88 eV, corresponding to S 2p, C 1s, N 1s, O 1s, and F 1s, respectively. The R-CDs contained 47.85% carbon, 10.99% nitrogen, 0.95% sulfur, 11.76% fluorine, and 28.44% oxygen. The high-resolution spectrum of the C 1s peak (Figure 1e) demonstrated three typically fitted peaks at 284.5, 285.8 and 288.4 eV, which were assigned to the C-C/C-S, C-N/C-O, and C-F bonds, respectively [50,51]. The high-resolution O 1s spectrum (Figure 1f) shows two peaks at 530.9 and 532.3 eV, which can be attributed to C=O and O=C-O bonds, respectively [52,53]. Three fitted peaks at binding energies of 399.1, 399.8 and 401.2 eV can be seen in the high-resolution N 1s spectrum (Figure 1g) [54], which correspond to pyridinic, amino and graphitic nitrogen atoms, respectively. In the high-resolution F 1s spectrum (Figure 1h), the XPS curve can be obtained from the curve fitting of two peaks at 685.2 and 686.9 eV, which are assigned to semi-ionic C-F and covalent C-F bonds, respectively. The high-resolution S 2p spectrum (Figure 1i) exhibited two peaks at 167.9 and 169.0 eV, which correspond to C-S and S-O bonds, respectively.

The absorption, excitation, and emission spectra of the R-CDs were extensively characterized using a spectrophotometer to achieve a comprehensive understanding of their optical properties. As depicted in Figure 2a, the UV-vis absorption spectrum of the R-CD solution exhibited an absorption peak at 263 nm, which corresponds to the n-π* transition of the C=O band [55]. The excitation and absorption spectra show an excitation peak centered at 462 nm and an emission peak centered at 504 nm. As shown in the inset of Figure 2a, irradiation with a 365 nm laser pen resulted in distinct yellow-green fluorescence. The 3D emission spectrum (Figure 2b) confirms that the R-CDs are typical bandgap-emitting R-CDs. The emission peak is independent of excitation and exhibits a unique maximum emission peak at 504 nm. To evaluate the potential of carbon dots in biological imaging and sensing applications, their fluorescence stability under various environmental conditions needs to be investigated. The fluorescence intensity was measured over time under controlled conditions. Figure 2c shows that the R-CDs exhibited excellent resistance to irradiation, maintaining a high fluorescence intensity even after 90 min of laser irradiation. As shown in Figure 2d, the carbon dots were highly stable over a wide pH range (pH = 2–11), with only a slight decrease in the fluorescence intensity observed at pH > 12. Notably, even under strongly acidic conditions (pH = 1), the carbon dots still maintained 72% of their original fluorescence intensity, indicating good fluorescence stability in acidic environments. Furthermore, Figure 2e shows that the R-CDs exhibited good salt concentration stability, with negligible fluorescence intensity changes observed at NaCl concentrations ranging from 0 to 1000 mg/mL. Finally, Figure 2f shows that the carbon dots maintained a high fluorescence intensity (>80%) even at temperatures as high as 90 °C, with only a slight decrease in fluorescence intensity observed with increasing temperature.

To investigate the catalytic effect of R-CDs, the catalytic behavior of R-CDs was analyzed through ESR spectroscopy and active species trapping experiments. Based on the visible light-driven properties of R-CDs, the ESR spectra were examined under visible light and dark conditions. Figure 3a–c shows the ESR spectra, which exhibited no signal response from the ROS in a dark environment. On the contrary, when TEMPO as the spin-trapping agent is illuminated by visible light (LED, 200 mW), the characteristic triplet signal (1:1:1) attributed to ^1^O_2_ can be clearly seen. Moreover, using DMPO as the spin-trapping agent can also confirming the existence of O_2_^−^ [56,57]. The chromogenic substrate TMB has a characteristic blue color when oxidized by ROS and can be used to detect the enzymatic activity of R-CDs. By analyzing the UV-absorption spectrum of TMB, characteristic absorption peaks at 325 and 651 nm were observed as the light exposure time increased owing to the greater noise at 370 nm compared to that at 651 nm. Therefore, the enzymatic activity of the R-CDs was determined by measuring the absorption of TMB at 651 nm. To determine the intermediates and products produced during the reaction, a series of trapping agents were used to remove or suppress particles that may affect the catalytic process. SOD can convert superoxide anions to oxygen and hydrogen peroxide [58]. EDTA, a commonly used chelating agent, inhibits the production of ROS associated with metal ions [59]. CuSO_4_, as an ion inhibitor, can effectively inhibit the free radicals produced by electron transfer [60]. TPA is a lipophilic compound that is primarily used to erase hydroxyl radicals (·OH) [61]. TRY, an amino acid oxidized by ROS, can effectively consume free radicals [62]. Figure 3d shows that the addition of SOD reduced the ROS activity to 60% compared to the control group, while the addition of CuSO_4_ reduced the ROS activity to 10%, and no significant changes were found in the other experimental groups. Based on the ESR spectra, it can be concluded that during the photocatalytic process, singlet oxygen and superoxide radicals are predominantly generated, with superoxide radicals being the main intermediates. Compared with organic catalysts, which can only work in relatively mild biological environments, inorganic catalysts have stronger enzyme activity stability. Therefore, we tested the ROS productivity of the R-CDs under various conditions. Figure 3e illustrates the effects of different temperatures on the ROS productivity of R-CDs, exhibiting an increase within a wide temperature range (20–50 °C), and the productivity reached the peak after 50 °C. Figure 3f compares the ROS productivity of the R-CDs in different pH environments, exhibiting stable characteristics in the pH range of 3–11. The experiment to determine the ROS productivity in NaCl solutions with different concentrations (Figure 3g) showed that the ROS concentration was slightly lower in NaCl solutions but maintained a high trend in strong ionic solutions. As shown in Figure 3h, the experimental group with Mn^2+^, Ca^2+^, Na^+^, Mg^2+^, Li^+^, Zn^2+^, and Ba^2+^ added to R-CDs/TMB solution had a negligible impact on the catalytic activity of R-CDs. Conversely, in solutions containing Cu^2+^ ions, the catalytic activity of the R-CDs was significantly inhibited owing to the inhibitory effect of Cu^2+^ on electron transfer. However, the occurrence of Cu^2+^ in typical bodies of water is relatively uncommon; therefore, the catalytic activity of R-CDs in most aquatic environments remains unaffected. Considering the potential use of R-CDs in natural aquatic environments, where they might enter the human body through the food chain, we conducted cytotoxicity tests using HeLa cancer cells as representatives. Since R-CDs cannot receive visible light inside the human body, the experiments were conducted in a dark environment. Figure 3i shows the experimental results, indicating that compared to the control group, cell viability remains above 80% even at the highest working concentration. These findings suggest that R-CDs can stably generate ROS in marine environments without significant harm to public health. However, further research is needed to investigate the long-term environmental impact and potential toxicity of R-CDs.

PGS is a microalgal species widely distributed in the ocean and is commonly associated with the occurrence of HABs. In this study, PGS was used as a representative model to investigate the anti-algal effects of R-CDs. PGS was cultured in a conical flask under light (LED, 5009 lux), and its growth curve was plotted based on daily UV-vis absorption at 650 nm (brown line in Figure 4a). As PGS reproduces, the color of the culture medium gradually changes to brown. Statistics show that PGS exhibited a maximum growth rate of 35% per day. To evaluate the anti-algal effects of R-CDs, PGS cells treated with R-CDs were labeled as the experimental group, and those without R-CDs were used as the control group. The experimental groups were exposed to various concentrations of R-CDs (0.3, 0.4, 0.5, and 0.6 mg/mL). Figure 4b illustrates that the addition of R-CDs led to a significant reduction in the PGS cell density over time. However, the anti-algal efficiency of different concentrations of R-CDs did not exhibit significant differences in practical use, which may be attributed to limited light power or saturation of the ROS concentration. Notably, a concentration of 0.4 mg/mL demonstrated the highest anti-algal efficacy. After 27 h of R-CD addition, the PGS cell density ceased to decrease, indicating complete algal death. In this study, the practical use of R-CDs as anti-algal agents was evaluated by adding PGS and R-CDs to water samples collected from the coast of Shenzhen Bay. The inactivation effect was evaluated by determining the relative inactivation rates. As shown in Figure 4c, the anti-algal effect in the real seawater group was better than that in the simulated water group. Indeed, R-CDs exhibit stronger efficacy in complex aquatic environments compared to laboratory conditions. Interestingly, after the light absorption rate of the real seawater group decreased to the lowest point, the absorbance increased again, indicating the low toxicity of the R-CDs to other marine organisms. Figure 4d depicts the working principle of R-CDs. The experimental group treated with R-CDs generated free radicals upon light exposure, which induced oxidative stress in the membrane structure of the PGS cells, leading to cell lysis and death. Consequently, the color of the liquid changed from brownish-yellow to clear. In summary, the photocatalytic R-CDs synthesized in this study exhibited rapid and stable anti-algal effects, rendering them suitable for use in various bodies of water susceptible to algal blooms. In addition, their low toxicity to non-target organisms makes them a promising anti-algal agent.

The anti-algal experiments demonstrated the excellent ability of the R-CDs to scavenge algae. To further confirm the effectiveness of R-CDs in killing algae, changes in PGS under the action of R-CDs were observed using SEM. Figure 5a shows images of PGS cells at 11,000× magnification. The PGS cells were elliptical in shape and approximately 1 μm in size. As the illumination time increased, changes in the cell surface morphology were observed; the cells changed from smooth and plump (0 and 2 h) to rough and wrinkled (4 h), and eventually the cells ruptured, appearing as hollow structures (6 h). Figure 5b shows images of the PGS cell clusters at 2000× magnification. Healthy PGS cells were uniformly dispersed, with only a small number of cells visible in the field of view (0 and 2 h). As the cells died, the accumulation of cell aggregates and lysis debris was observed (4 h). As the cells continued to die, PGS aggregates accumulated in the field of view (6 h). These results suggest that R-CDs kill PGS by cleaving the membrane structure of PGS cells, leading to the aggregation of cell residues.

The SEM images indicate that R-CDs cause the lysis and death of PGS cells. To further verify that R-CDs produce ROS within cells, CLSM (confocal laser scanning microscopy) imaging was used to examine the distribution of ROS and R-CDs inside PGS cells. As shown in Figure 6a, in the Ex488 channel, PGS showed almost no autofluorescence. Following the co-incubation of R-CDs with PGS for 30 min and subsequent washing with PBS, CLSM images of the cells were obtained (Figure 6b). Fluorescence signals were clearly detected within the PGS cells, indicating successful internalization of the R-CDs. Finally, we used the ROS-Brite 670 probe, which can produce bright near-infrared fluorescence after reacting with ROS, to verify whether the R-CDs generated ROS. The CLSM image of R-CDs + ROS-Brite 670 (Figure 6c) shows that the fluorescence area of the Ex488 channel and Ex640 channel highly overlap. The results of this investigation demonstrate that R-CDs can penetrate the membrane of algal cells and subsequently generate ROS within their intracellular environment.

Excessive ROS can trigger oxidative stress, causing the cell membrane structure to release ROS scavengers and reduce the ROS content. FLIM technology was employed in this study for mechanistic analysis. While the CLSM image (Figure 7a) provides less obvious information about cell morphology, the FLIM image (Figure 7b) reveals changes in the fluorescence lifetime in addition to cell shape. With no illumination, the overall fluorescence lifetime of the cells was relatively uniform, with an average lifetime of 319.2 ps. Furthermore, a disparity in fluorescence lifetime was observed between the central region and the periphery of the cell. Following light exposure, there was a significant decrease in the average fluorescence lifetime, suggesting alterations in the physiological activity of the subcellular structures within the PGS cells. As depicted in Figure 7c, with increasing illumination time, the average fluorescence lifetime initially exhibited a significant decrease. However, as the duration of illumination increased, the fluorescence lifetime gradually returned to its original level. This result suggests that oxidative stress caused by ROS production by R-CDs under light exposure causes a drastic change in the physiological state of PGS. The morphology of the fluorescence lifetime fitting curve in Figure 7d did not change, indicating that the R-CDs themselves remained unchanged and functioned solely as catalysts.

Using the phase diagram shown in Figure 8a, the binding of R-CDs to the external (I) and internal (II) subcellular structures of the PGS cells can be differentiated. As is shown in Figure 8b, internal and external parts of the combination of R-CDs and PSG cells have different lifetimes. Upon light activation, the R-CDs generated ROS and triggered oxidative stress (Figure 8c,d). Ribosomes in the endoplasmic reticulum released ROS scavengers, leading to a decrease in the fluorescence lifetime of R-CDs. As shown in Figure 8e, a more significant change in the lifespan occurred because of the higher concentration of ROS scavengers surrounding the endoplasmic reticulum. With increasing exposure time, the oxidative stress defense line weakened, and ROS scavengers decomposed, causing the fluorescence lifetime to return to its original level, as illustrated in Figure 8f. This is consistent with the reported changes in the levels of antioxidant detoxifying agents during oxidative stress-induced cellular apoptosis [63]. The use of R-CDs combined with FLIM can be an effective research method.

## 3. Methods and Materials

### 3.1. Instruments and Reagents

A variety techniques were employed to analyze the materials under investigation. These included TEM utilizing an FEI Tecnai G2 F20 instrument (Hillsboro, OR, USA), XPS utilizing a Thermo Fisher ESCALAB instrument (Waltham, MA, USA), FT-IR utilizing a Nicolet 5700 spectrometer (Waltham, MA, USA), and UV-Vis utilizing a UV-2550 Shimadzu instrument (Kyoto, Japan). The electron paramagnetic resonance (EPR) data of CDs were recorded using Bruker EMX Plus (Billerica, MA, USA), fluorescence spectroscopy employing a Fluorlog^@_3^ Steady-State spectrofluorometer from HORIBA Scientific (Kyoto, Japan). Confocal fluorescence (CLSM) and fluorescence lifetime image (FLIM) utilized laser-scanning confocal fluorescence microscopy (Nikon, A1R MP +and Carl Zeiss, LSM 800with Airyscan) and Leica SP8 confocal microscopy. Analytical-grade 2,4-difluorobenzoic acid, lysine, benzene sulfonamide, and 3, 3′, 5, 5′-tetramethylbenzidine (TMB), as well as ethanol and metal salts (MnCl_2_, CaCl_2_, CuSO_4_, NaCl, MgCl_2_·6H_2_O, LiCl, ZnCl_2_, BaCl_2_), were purchased from the Macklin Biochemical Co. Ltd. (Shanghai, China). All solvents were prepared using ultrapure water. CCK-8 and PBS solutions were purchased from the GIBCO Thermo Fisher Scientific Co., Ltd. (Shanghai, China). The ROS Brite^TM^ 670 was purchased from AAT Bioquest Inc. (Pleasanton, CA, USA). The SOD activity detection kit was purchased from the Solarbio Science and Technology Co., Ltd. (Beijing, China). PGS was obtained from the Kirgen Bioscience Co., Ltd. (Shanghai, China). The F/2 medium and sea salt was purchased from the Shanghai Guangyu Biological Technology Co., Ltd. (Shanghai, China). Phosphate buffered saline was purchased from Biosharp, China.

### 3.2. Preparation of R-CDs

R-CDs were synthesized via the hydrothermal method in two steps. First, 2,4-difuorbenzoic acid (1 g) and lysine (0.637 g) were dissolved in deionized water (25 mL), and ethanol (2 mL) was added to assist dissolution in a Teflon-lined autoclave (50 mL), followed by heating at 180 °C for 24 h. After cooling to room temperature, the solution was centrifuged and filtered to obtain clear orange solution A. Second, solution A (3 mL) and benzene sulfonamide (3 mM) were added to a mixture of deionized water (25 mL) and ethanol (2 mL), followed by heating at 180 °C for 8 h. After cooling, filtration was performed to obtain a clear yellow-green R-CD solution. The R-CD solution was freeze-dried to obtain R-CD powder for subsequent weighing and characterization. The concentration of R-CDs was determined to be 6.25 mg/mL and was diluted to 4 mg/mL with deionized water for subsequent experiments.

### 3.3. PGS Scherffel Culture

Algae were cultured with different concentrations of R-CDs (0, 3, 4, 5, and 6 mg/mL) in a simulated natural light culture environment. Shaking (twice per day) was performed to ensure algal dispersion and improve the growth rate. The cell density of the algae was monitored using a UV-vis spectrophotometer spectrofluorometer. Algal growth was reflected by measuring the ultraviolet absorbance of the sample at 650 nm (OD650).

### 3.4. Determination of the ROS Content

#### 3.4.1. Determination of the ROS Content

The color reaction between TMB and ROS reflects the ROS productivity of R-CDs [64]. The intensity of the infrared absorbance spectrum (OD652) was used to characterize the ROS content; the higher the intensity, the higher the content. A 1 mL detection solution will be prepared using PBS buffer to mix R-CDs and TMB, with a concentration of 0.4 mg/mL for R-CDs and 4 mM for TMB. The absorbance will be measured using UV-Vis spectroscopy after exposure to light.

#### 3.4.2. Stability Experiment

Experimental investigations were conducted to assess the photocatalytic stability of R-CDs (0.4 mg/mL) under various conditions, encompassing different temperatures (20–80 °C), buffered solutions with varying pH values (pH = 3–11), different NaCl concentrations (0–50 mg/mL), and solutions containing different ions (25 μM of Mn^2+^, Ca^2+^, Na^+^, Mg^2+^, Li^+^, Zn^2+^, and Ba^2+^). The R-CDs/TMB mixture dissolved in ultrapure water at 20 °C served as the control group. Following 10 min of visible light irradiation (LED, 5000 lux), OD652 was measured to establish A_0_. The experimental group's OD652, denoted as A, was measured after controlling variables. The ratio A/A_0_ was computed, and the entire process was repeated three times for accuracy.

### 3.5. Anti-Algae Experiment

A simulated artificial seawater solution was prepared by dissolving sea salt in ultrapure water to achieve a concentration of 33 g/L. To ensure sterility and eliminate microbial interference, the artificial seawater was sterilized using high-temperature steam sterilization. Following sterilization, the seawater was prepared according to the formula of F/2 medium. This involved adding 0.00565 g/L NaH_2_PO_4_·H_2_O, 0.075 g/L NaNO_3_, 1 mL/L trace metal solution, and 1 mL/L vitamin solution through a 0.22 μm microporous membrane. PSG was inoculated into the culture medium and cultivated in a growth chamber at 25 °C under a mixed three-color light simulating natural sunlight (LED, 5000 lux), with a light–dark cycle of 12 h:12 h. The algae population density was characterized by measuring the absorbance of PSG at 652 nm, and the growth curve was plotted. Different concentrations of R-CDs (0.0, 0.3, 0.4, 0.5, 0.6 mg/mL) were employed to treat the PSG algae population. Equal concentrations of R-CDs solution were used as a background to remove interference. The time-dependent change in OD652 was measured to represent changes in algae concentration, and corresponding growth inhibition curves were generated. Furthermore, the anti-algae performance of R-CDs was evaluated in real water samples. Algal suspension was separated through low-speed centrifugation (2000 rpm) from the culture medium. For the real seawater group, a portion of the algal suspension was mixed with water samples collected from Shenzhen Bay, while an equal amount of culture medium was added to the simulated seawater group. R-CDs were introduced to achieve a concentration of 0.4 mg/mL. The corresponding culture medium/R-CDs mixture was used as a background to measure the time-dependent changes in OD652 for both the real seawater and simulated seawater groups, and growth inhibition curves were plotted.

### 3.6. Imaging Experience

#### 3.6.1. TEM Imaging

After sampling R-CDs, they were subjected to ultrasonic treatment for 10 mins in a centrifuge tube. The post-sonication R-CDs solution was then dripped onto a microgrid copper mesh and placed in an oven for drying at 60 °C. Lastly, the prepared sample was observed using a TEM instrument under a voltage of 120 kV.

#### 3.6.2. SEM Sample Preparation of Algae

The employment of a gradient dehydration approach in processing PSG enables the preservation of its cellular morphology to the greatest extent possible, preventing disruption due to stress-induced reactions [65]. PSG cells were extracted from the culture medium and incubated with R-CDs (0.4 mg/mL). Samples were collected after 0, 2, 4, and 6 h of exposure under light. Following centrifugation at 4000 rpm for 5 min, the cells were immersed in a 2.5% glutaraldehyde solution for 4 h. Subsequently, under the same centrifugation conditions, the cells underwent sequential dehydration steps involving immersion in ethanol solutions with concentrations of 30%, 45%, 60%, 75%, 90%, and finally 100% for 5 min each, followed by centrifugation. Subsequently, the algae were carefully deposited onto clean silicon wafers and dried in a bake-out furnace at 60 °C for 5 days. Lastly, a gold coating was applied to the silicon wafer surface, and the samples were subjected to SEM for morphological analysis.

#### 3.6.3. CLSM and FLIM Sample Preparation of Algae

CLSM was employed to verify the ROS generation effect of R-CDs within cells. Three samples of algal suspension were extracted from the culture medium. One of these samples, incubated directly under light exposure (LED, 5000 lux) for 30 min, was designated as the control group. The other two samples, after the addition of R-CDs (0.4 mg/mL), underwent incubation under identical conditions and were labeled as the R-CDs group. Upon completion of incubation, centrifugation was performed (8000 rpm, 3 min), followed by washing with PBS to remove excess R-CDs and culture medium impurities. This process was repeated three times. Subsequently, one of the precipitates from the R-CDs group was combined with an ROS detection probe (ROS BriteTM 670) and incubated for 15 min. Following this, the washing procedure was repeated. This sample was denoted as the R-CDs + ROS BriteTM 670 group [66]. The prepared suspension was used to create CLSM observation slides, with images recorded under bright-field illumination, as well as under 488 and 640 nm laser excitation.

To observe the fluorescence lifetime variation of R-CDs within algal cells using FLIM, R-CDs (0.4 mg/mL) were incubated with PSG under light-avoiding conditions in a dark environment for 30 min. Subsequently, the samples were washed and prepared in the same manner as the CLSM samples. The prepared samples were positioned on the FLIM microscope stage, and microscope light sources were employed to induce ROS generation in R-CDs. Fluorescence lifetime images of R-CDs were then recorded at sequential time intervals (0, 5, 10, 15, 20, 25 min) under illumination (excitation 504 nm). Finally, SPC Image Version 9.88 (Becker Hickl) software was utilized for the analysis of FLIM imaging data.

## 4. Conclusions

In this study, we successfully synthesized ROS-generating R-CDs using a simple hydrothermal method. The anti-algal effects of these R-CDs were validated through experimental investigations. The use of SEM, CLSM, and FLIM techniques offers valuable insights into the intricate mechanisms underlying the mode of action of anti-algal agents. The results demonstrate that the photocatalytic activity of R-CDs led to the generation of abundant ROS, which effectively disrupted the cell membrane of PGS, resulting in cell lysis and death. We believe that this work would enhance our comprehensive understanding of the effectiveness of similar agents in combating algal growth. 

## Data Availability

The data presented in this study are available in article.

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
