# Peer review of "Visible-Light-Activated Carbon Dot Photocatalyst for ROS-Mediated Inhibition of Algae Growth"

_ijms, 2023, doi:10.3390/ijms241713509_

Round 1
Reviewer 1 Report
1. Introduction does not contain sufficient data on carbon dots as unique photosenzitizers. I kindly ask authors to write separate paragraph on carbon dots with emphasis on their potential to generate reactive oxygen species. At least 10 papers should be cited in this section such as :1. Nanomaterials 2022, 12, 4070. https://doi.org/10.3390/nano12224070 and 2. Pharmaceutics 2023, 15, 1170. https://doi.org/10.3390/pharmaceutics15041170
2. During preparation of carbon dots, authors used as precursors 2,4-difluorobenzoic acid, lysine, benzene sulfonamide . These precursors are much more costly than citric acid or other precursors for carbon dots preparation. Aim of the paper is to prepare agent for large scale algae eradication and for this purpose carbon dots should be as cheaper as possible. Please explain what are advantages of using exactly these precursors mentioned in the manuscript.
3. Section 2. Experimental does not contain details on sample preparation for EPR measurement. What light source was used to irradiate samples? Please write it. Also there are no details on sample preparation for TEM measurement. What type of grid was used? What was the voltage on TEM? Please write it. What light source, power, wavelength was used to initiate algae eradication (text on page 8, form line 230). There is not a single word about Confocal laser scanning microscopy in Experimental. What instrument was used? What light source was used for excitation? How did you prepare samples? Please write details on FLIM apparatus.
4. Fig 2a shows that absorption of sample in visible region is equal to zero. I kindly ask authors to explain how solar light can trigger carbon dots to produce ROS and annihilate algae? This result indicate that only UV light cant trigger ROS generation.
5. I kindly ask authors to confirm singlet oxygen and OH production using FLUOROLOG and UV VIS. Use sensor green, terephtalic acid as probes on Flurolog and ABDA, DBPF as probes on UV VIS. The best option is to dissolve R-CD in deuterated acetone or chloroform and measure luminescence at 1270 nm. This is best evidence of singlet oxygen generation.
Minor editing of English is required. Certain letters are missing in words.
Author Response
Response to the Referees’ Comments
We thank the referees’ positive and valuable comments. The following are point-by-point to the referees’ comments/questions.
Reviewer #1:
1.Introduction does not contain sufficient data on carbon dots as unique photosenzitizers. I kindly ask authors to write separate paragraph on carbon dots with emphasis on their potential to generate reactive oxygen species. At least 10 papers should be cited in this section such as :1. Nanomaterials 2022, 12, 4070. https://doi.org/10.3390/nano12224070 and 2. Pharmaceutics 2023, 15, 1170. https://doi.org/10.3390/pharmaceutics15041170
Response 1:
We do appreciate your positive and valuable comments. In the "Introduction" section, we have rephrased and supplemented the content regarding the potential of carbon dots to generate ROS, referencing the citations you recommended. The revised portion is presented below:
CDs, a class of non-metallic nanomaterials, have attracted significant attention due to their unique properties, including tunable fluorescence, excellent stability, high photoactivity, and low toxicity [33]. Their applications span from photocatalysis and sensitive sensing to advanced biological imaging [34-40]. As photocatalysts, carbon dots, including graphene quantum dots (GQDs), carbon quantum dots (CQDs), carbon nanodots (CNDs), and carbonized polymer dots (CPDs), have attracted much attention in the field of photo fungicides due to their ability to generate ROS under light, good biocompatibility, good solubility in water, and light bleaching resistance [41-43]. Reports of using photodynamic therapy to eliminate harmful units such as bacterial cells and cancer cells are often mentioned [44-47]. However, the utilization of CDs as non-metallic photocatalyst for anti-algal applications has been rarely reported. On the other hand, research and development on the oxidative stress process of algae are relatively lagging behind
Reference
[33] S. J. Zhu, Y. B. Song, X. H. Zhao, J. R. Shao, J. H. Zhang, B. Yang, The photoluminescence mechanism in carbon dots (graphene quantum dots, carbon nanodots, and polymer dots): current state and future perspective. Nano Research 8, 2(2015) 355-381.
[34] J. Jin, L. L. Li, L. H. Zhang, Z. H. Luan, S. Q. Xin, K. Song, Progress in the Application of Carbon Dots-Based Nanozymes. Front. Chem. 9, (2021) 8.
[35] Y. F. Liu, X. S. Tang, T. Zhu, M. Deng, I. P. Ikechukwu, W. Huang, G. L. Yin, Y. Z. Bai, D. R. Qu, X. B. Huang, F. Qiu, All-inorganic CsPbBr3 perovskite quantum dots as a photoluminescent probe for ultrasensitive Cu2+ detection. Journal of Materials Chemistry C 6, 17(2018) 4793-4799.
[36] M. Du, B. L. Huo, J. M. Liu, M. W. Li, A. Shen, X. Bai, Y. R. Lai, L. Q. Fang, Y. X. Yang, A turn-on fluorescent probe based on Si-rhodamine for sensitive and selective detection of phosgene in solution and in the gas phase. Journal of Materials Chemistry C 6, 39(2018) 10472-10479.
[37] G. K. Wang, S. L. Wang, C. L. Yan, G. Y. Bai, Y. F. Liu, DNA-functionalized gold nanoparticle-based fluorescence polarization for the sensitive detection of silver ions. Colloids and Surfaces B-Biointerfaces 167, (2018) 150-155.
[38] S. P. Jackson, J. Bartek, The DNA-damage response in human biology and disease. Nature 461, 7267(2009) 1071-1078.
[39] Y. Q. Dong, J. W. Shao, C. Q. Chen, H. Li, R. X. Wang, Y. W. Chi, X. M. Lin, G. N. Chen, Blue luminescent graphene quantum dots and graphene oxide prepared by tuning the carbonization degree of citric acid. Carbon 50, 12(2012) 4738-4743.
[40] Z. Q. Guo, G. H. Kim, I. Shin, J. Yoon, A cyanine-based fluorescent sensor for detecting endogenous zinc ions in live cells and organisms. Biomaterials 33, 31(2012) 7818-7827.
[41] S. Jovanovic, Z. Markovic, M. Budimir, J. Prekodravac, D. Zmejkoski, D. Kepic, A. Bonasera, B. T. Markovic, Lights and Dots toward Therapy-Carbon-Based Quantum Dots as New Agents for Photodynamic Therapy. Pharmaceutics 15, 4(2023).
[42] A. Baranwal, A. Srivastava, P. Kumar, V. K. Bajpai, P. K. Maurya, P. Chandra, Prospects of Nanostructure Materials and Their Composites as Antimicrobial Agents. Frontiers in Microbiology 9, (2018).
[43] C. Sakdaronnarong, A. Sangjan, S. Boonsith, D. C. Kim, H. S. Shin, Recent Developments in Synthesis and Photocatalytic Applications of Carbon Dots. Catalysts 10, 3(2020).
[44] Z. M. Markovic, M. Kovacova, S. R. Jeremic, S. Nagy, D. D. Milivojevic, P. Kubat, A. Kleinova, M. D. Budimir, M. M. Mojsin, M. J. Stevanovic, A. Annusova, Z. Spitalsky, B. M. T. Markovic, Highly Efficient Antibacterial Polymer Composites Based on Hydrophobic Riboflavin Carbon Polymerized Dots. Nanomaterials 12, 22(2022).
[45] V. Ahuja, S. Banerjee, P. Roy, A. K. Bhatt, Fluorescent xylitol carbon dots: A potent antimicrobial agent and drug carrier. Biotechnology and Applied Biochemistry 69, 4(2022) 1679-1689.
[46] Z. Y. Wang, L. A. Sheng, X. X. Yang, J. D. Sun, Y. L. Ye, S. X. Geng, D. L. Ning, J. Y. Zheng, M. H. Fan, Y. Z. Zhang, X. L. Sun, Natural biomass-derived carbon dots as potent antimicrobial agents against multidrug-resistant bacteria and their biofilms. Sustainable Materials and Technologies 36, (2023).
[47] X. L. Dong, W. X. Liang, M. J. Meziani, Y. P. Sun, L. J. Yang, Carbon Dots as Potent Antimicrobial Agents. Theranostics 10, 2(2020) 671-686.
- During preparation of carbon dots, authors used as precursors 2,4-difluorobenzoic acid, lysine, benzene sulfonamide. These precursors are much more costly than citric acid or other precursors for carbon dots preparation. Aim of the paper is to prepare agent for large scale algae eradication and for this purpose carbon dots should be as cheaper as possible. Please explain what are advantages of using exactly these precursors mentioned in the manuscript.
Response 2:
Thanks for your careful review! Indeed, the materials needed for the synthesis of carbon dots are quite minimal. Furthermore, the concentration required for efficient algal eradication is remarkably low, and these carbon dots exhibit a sustained action without self-depletion, thereby ensuring prolonged effectiveness. Hence, considering all these factors, the overall cost remains relatively low.
- 3. Section 2. Experimental does not contain details on sample preparation for EPR measurement. What light source was used to irradiate samples? Please write it. Also there are no details on sample preparation for TEM measurement. What type of grid was used? What was the voltage on TEM? Please write it. What light source, power, wavelength was used to initiate algae eradication (text on page 8, form line 230). There is not a single word about Confocal laser scanning microscopy in Experimental. What instrument was used? What light source was used for excitation? How did you prepare samples? Please write details on FLIM apparatus.
Response 3:
Thanks for your careful and valuable comments. We deeply recognize the presence of omissions in the experimental description section. In the new submission, we have supplemented this content. The following is provided:
- Instruments and reagents
A variety techniques were employed to analyze the materials under investigation. These included TEM utilizing a FEI Tecnai G2 F20 instrument (USA), XPS utilizing a Thermo Fisher ESCALAB instrument (USA), FT-IR utilizing a Nicolet 5700 spectrometer (USA), UV-Vis utilizing a UV-2550 Shimadzu instrument (Kyoto, Japan), the electron paramagnetic resonance (EPR) data of CDs were recorded using Bruker EMX Plus, fluorescence spectroscopy employing a Fluorlog@_3 Steady-State spectrofluorometer from HORIBA Scientific (Japan), electron spin-resonance spectroscopy utilizing a BRUKER EMX Plus instrument (Germany), confocal fluorescence (CLSM) and fluorescence lifetime image (FLIM) was utilizing Laser-scanning confocal fluorescence microscopy (Nikon, A1R MP +and Carl Zeiss, LSM 800with Airyscan) and Leica SP8 confocal microscopy. Analytical grade 2,4-difluorobenzoic acid, lysine, benzene sulfonamide, and 3, 3’, 5, 5’-tetramethylbenzidine (TMB) as well as ethanol and metal salts (MnCl2, CaCl2, CuSO4, NaCl, MgCl2ž6H2O, LiCl, ZnCl2, BaCl2) were purchased from the Macklin Biochemical Co. Ltd. (Shanghai, China). All solvents were prepared using ultrapure water. CCK-8 and PBS solutions were purchased from the GIBCO Thermo Fisher Scientific Co., Ltd. (Shanghai, China). The ROS BriteTM 670 was purchased from AAT Bioquest Inc., (USA). The SOD activity detection kit was purchased from the Solarbio Science and Technology Co., Ltd. (Beijing, China). PGS was obtained from the Kirgen Bioscience Co., Ltd. (Shanghai, China). The F/2 medium and sea salt was purchased from the Shanghai Guangyu Biological Technology Co., Ltd., (Shanghai, China). Phosphate buffered saline was purchased from Biosharp, China.
2.2 Preparation of R-CDs
R-CDs were synthesized via the hydrothermal method in two steps. First, 2,4-difuorbenzoic acid (1 g) and lysine (0.637 g) were dissolved in deionized water (25 mL), and ethanol (2 mL) was added to assist dissolution in a Teflon-lined autoclave (50 mL), followed by heating at 180 °C for 24 h. After cooling to room temperature, the solution was centrifuged and filtered to obtain clear orange solution A. Second, solution A (3 mL) and benzene sulfonamide (3 mmol) were added to a mixture of deionized water (25 mL) and ethanol (2 mL), followed by heating at 180 °C for 8 h. After cooling, filtration was performed to obtain a clear yellow–green R-CD solution. The R-CD solution was freeze-dried to obtain R-CD powder for subsequent weighing and characterization. The concentration of R-CDs was determined to be 6.25 mg/mL and was diluted to 4 mg/mL with deionized water for subsequent experiments.
2.3 PGS Scherffel culture
Algae were cultured with different concentrations of R-CDs (0, 3, 4, 5, and 6 mg/mL) in a simulated natural light culture environment. Shaking (twice per day) was performed to ensure algal dispersion and improve the growth rate. The cell density of the algae was monitored using a UV-vis spectrophotometer spectrofluorometer. Algal growth was reflected by measuring the ultraviolet absorbance of the sample at 650 nm (OD650).
2.4 Oxidase-like activity assessment
2.4.1 Determination of the ROS content
The color reaction between TMB and ROS reflects the ROS productivity of R-CDs [48]. The intensity of the infrared absorbance spectrum (OD652) was used to characterize the ROS content; the higher the intensity, the higher the content. A 1 ml detection solution will be prepared using PBS buffer to mix R-CDs and TMB, with a concentration of 0.4 mg/ml for R-CDs and 4 mM for TMB. The absorbance will be measured using UV-Vis spectroscopy after exposure to light.
2.4.2 Stability experiment
Experimental investigations were conducted to assess the photocatalytic stability of R-CDs (0.4 mg/ml) under various conditions, encompassing different temperatures (20-80 °C), buffered solutions with varying pH values (pH = 3-11), different NaCl concentrations (0-50 mg/ml), and solutions containing different ions (25 μM of Mn2+, Ca2+, Na+, Mg2+, Li+, Zn2+, and Ba2+). The R-CDs/TMB mixture dissolved in ultrapure water at 20°C served as the control group. Following 10 minutes of visible light irradiation (LED, 5000 lux), OD652 was measured to establish A0. The experimental group's OD652, denoted as A, was measured after controlling variables. The ratio A/A0 was computed, and the entire process was repeated three times for accuracy.
2.5 Anti-algae experiment
A simulated artificial seawater solution was prepared by dissolving sea salt in ultrapure water to achieve a concentration of 33 g/L. To ensure sterility and eliminate microbial interference, the artificial seawater was sterilized using high-temperature steam sterilization. Following sterilization, the seawater was prepared according to the formula of F/2 medium. This involved adding 0.00565 g/L NaH2PO4·H2O, 0.075 g/L NaNO3, 1 ml/L trace metal solution, and 1 ml/L vitamin solution through a 0.22 μm microporous membrane. PSG was inoculated into the culture medium and cultivated in a growth chamber at 25 °C under a mixed three-color light simulating natural sunlight (LED, 5000 lux), with a light-dark cycle of 12h:12h. The algae population density was characterized by measuring the absorbance of PSG at 652 nm, and the growth curve was plotted. Different concentrations of R-CDs (0.0, 0.3, 0.4, 0.5, 0.6 mg/ml) were employed to treat the PSG algae population. Equal concentrations of R-CDs solution were used as a background to remove interference. The time-dependent change in OD652 was measured to represent changes in algae concentration, and corresponding growth inhibition curves were generated. Furthermore, the anti-algae performance of R-CDs was evaluated in real water samples. Algal suspension was separated through low-speed centrifugation (2000 rpm) from the culture medium. For the real seawater group, a portion of the algal suspension was mixed with water samples collected from Shenzhen Bay, while an equal amount of culture medium was added to the simulated seawater group. R-CDs were introduced to achieve a concentration of 0.4 mg/ml. The corresponding culture medium/R-CDs mixture was used as a background to measure the time-dependent changes in OD652 for both the real seawater and simulated seawater groups, and growth inhibition curves were plotted.
2.6 Imaging experience
2.6.1 TEM imaging
After sampling R-CDs, they were subjected to ultrasonic treatment for 10 mins in a centrifuge tube. The post-sonication R-CDs solution was then dripped onto a microgrid copper mesh and placed in an oven for drying at 60°C. Lastly, the prepared sample was observed using a TEM instrument under a voltage of 120 kV.
2.6.2 SEM sample preparation of algae
The employment of a gradient dehydration approach in processing PSG enables the preservation of its cellular morphology to the greatest extent possible, preventing disruption due to stress-induced reactions [49]. PSG cells were extracted from the culture medium and incubated with R-CDs (0.4 mg/ml). Samples were collected after 0, 2, 4, and 6 hours of exposure under light. Following centrifugation at 4000 rpm for 5 minutes, the cells were immersed in a 2.5% glutaraldehyde solution for 4 hours. Subsequently, under the same centrifugation conditions, the cells underwent sequential dehydration steps involving immersion in ethanol solutions with concentrations of 30%, 45%, 60%, 75%, 90%, and finally 100% for 5 minutes each, followed by centrifugation. Subsequently, the algae were carefully deposited onto clean silicon wafers and dried in an bake out furnace at 60°C for 5 days. Lastly, a gold coating was applied to the silicon wafer surface, and the samples were subjected to SEM for morphological analysis.
2.6.3 CLSM and FLIM sample preparation of algae
CLSM was employed to verify the ROS generation effect of R-CDs within cells. Three samples of algal suspension were extracted from the culture medium. One of these samples, incubated directly under light exposure (LED, 5000 lux) for 30 minutes, was designated as the control group. The other two samples, after the addition of R-CDs (0.4 mg/ml), underwent incubation under identical conditions and were labeled as the R-CDs group. Upon completion of incubation, centrifugation was performed (8000 rpm, 3 minutes), followed by washing with PBS to remove excess R-CDs and culture medium impurities. This process was repeated three times. Subsequently, one of the precipitates from the R-CDs group was combined with a ROS detection probe (ROS BriteTM 670) and incubated for 15 minutes. Following this, the washing procedure was repeated. This sample was denoted as the R-CDs + ROS BriteTM 670 group [50]. The prepared suspension was used to create CLSM observation slides, with images recorded under bright-field illumination, as well as under 488nm and 640nm laser excitation.
To observe the fluorescence lifetime variation of R-CDs within algal cells using FLIM, R-CDs (0.4 mg/ml) were incubated with PSG under light-avoiding conditions in a dark environment for 30 minutes. Subsequently, the samples were washed and prepared in the same manner as the CLSM samples. The prepared samples were positioned on the FLIM microscope stage, and microscope light sources were employed to induce ROS generation in R-CDs. Fluorescence lifetime images of R-CDs were then recorded at sequential time intervals (0, 5, 10, 15, 20, 25 mins) under illumination (excitation 504nm). Finally, SPC Image (Becker & Hickl) software was utilized for the analysis of FLIM imaging data.
Reference
[48] X. H. Chen, C. Zhang, L. J. Tan, J. T. Wang, Toxicity of Co nanoparticles on three species of marine microalgae. Environmental Pollution 236, (2018) 454-461.
[49] N. Gao, J. Jing, H. Z. Zhao, Y. Z. Liu, C. L. Yang, M. X. Gao, B. K. Chen, R. B. Zhang, X. L. Zhang, Defective Ag-In-S/ZnS quantum dots: an oxygen-derived free radical scavenger for mitigating macrophage inflammation. Journal of Materials Chemistry B 9, 43(2021) 8971-8979.
- Fig 2a shows that absorption of sample in visible region is equal to zero. I kindly ask authors to explain how solar light can trigger carbon dots to produce ROS and annihilate algae? This result indicate that only UV light cant trigger ROS generation.
Response 4:
Thanks for your careful and positive comments. Indeed, R-CDs exhibit a certain degree of absorption in the visible light range of 390-520nm, which might not be readily apparent from Fig. 2a. We apologize for any misunderstanding caused. Below, we provide an enlarged view of the absorption curve for your reference. This result, combined with the experimental observations, highlights that the generation of ROS by R-CDs requires only a minimal light source.
- 5. I kindly ask authors to confirm singlet oxygen and OH production using FLUOROLOG and UV VIS. Use sensor green, terephtalic acid as probes on Flurolog and ABDA, DBPF as probes on UV VIS. The best option is to dissolve R-CD in deuterated acetone or chloroform and measure luminescence at 1270 nm. This is best evidence of singlet oxygen generation.
Response 5:
Thank you very much for your kind and valuable feedback. Your rigorous requirements for the experiments are truly commendable. However, we believe that the experiments conducted through ESR and ROS scavenger assays have already confirmed the type of ROS. Considering the limited length of the article and the relatively lower significance of this experimental aspect, we find it more appropriate not to further expand on this particular section.

Reviewer 2 Report
Despite the article “Visible-Light-Activated Carbon Dot Nanoenzymes for ROS-Mediated Inhibition of Algae Growth” reveals interesting research ideas, and massive number of experiments both in the physical and chemical characterization and on the biological tests, I perceived unacceptable poor attention on the manuscript preparation. All the text is deficient of detailed descriptions of the experiments and lacks literature references supporting. It sounds very approximate on the discussion of the experimental results and on the description of graphs and images. I would like to consider the best scenario suggesting an extensive revision of the text after the publication on this journal.
Some comments are below reported:
· In the great part of the text, starting with the title, the enzymatic or nanoenzymatic activity of R-CDs is suggested. I consider inappropriate the use of this definition because it is universally attributed to biological entities – such as proteins or nucleic acids – catalysing biochemical reactions. I suggest the use of photocatalytic or catalytic activity terms to define the chemical behaviour of R-CDs when producing ROS under light stimulation.
· Sect. 2.2 – Centrifugation speed, time and instrument type are not depicted (line 91). On the end of the paragraph (lines 96-97), it is written that a “4 mg/mL dispersion in deionized water of R-CDs is used for subsequent experiments”, but it is not specified which type. Note that biological experiments were performed on completely different concentrations.
· Line 105 – Please, make sure that abbreviations have been correctly deciphered (TBN).
· Line 106 – It is reported that “the intensity of the infrared absorption spectrum (OD 652) was used to characterize the ROS content”. TBN assay for ROS content quantification is colorimetric, exploiting the UV/Vis light absorption.
· Lines 112-113 – Please, add the standard deviation value for the average size of R-CDs.
· When attributing infrared absorption and XPS peaks some relevant references must be cited. Moreover, to the unambiguously FT-IR peaks attributions it is necessary to show the complete spectrum that is truncated around 1550 cm-1 in Fig. 1c. Looking at XPS O 1s high resolution spectrum O=C-O and C=O contributions are reported, but it seems that no typical absorptions of this functional groups are present on the FT-IR spectrum. How the authors explain this apparent contraddiction?
· Line 142 – Add spectrofluorometer next to “spectrophotometer”.
· Line 143 – Add UV/Vis after “absorption spectrum”.
· Lines 144-145 – Add a reference for the 263 nm peak attribution.
· An emission band centred around 504 nm is reported for R-CDs, but the excitation wavelength is missing and more importantly, the authors found that the excitation spectrum is centred around 462 nm but, again, the emission wavelength is missing. In addition, how the authors explains that the excitation spectrum does not match with the UV/Vis absorption spectrum?
· Lines 153-156 – Please specify light source type and its intensity used for R-CDs irradiation and R-CDs concentration. Excitation and emission wavelengths must be depicted. The same considerations must be extended to pH stability.
· Line 161 – NaCl concentration is incorrectly reported on mM/mL, please be careful.
· Please, report the irradiation light source (type, intensity, experimental setup inside EPR spectrometer) employed for EPR experiments. Reinforce the results with literature references.
· I consider necessary adding, maybe in supporting information file, data (spectra, graphs) coming from TMB assays including calibration curves. There must be present also the control experiments performed with SOD, EDTA, CuSO4, TPA, TRY.
· There is no trace on the discussion about the experiments performed by EPR with DMPO trapping agent. Revise it.
· It is not clear which type of assay was utilized for the ROS production experiments in Fig. 3e, 3f, 3g, 3h, 3i. Moreover, neither the concentration of the various metal cations none the ionic strength of the solution is reported. In line 206 it is mentioned “plasma”, what the authors intended to say? In addition, who are the “non-target organisms in aquatic environment” (lake and marine) for the experiments in Fig. 3i? I can’t see any trace of discussion about the latter experiment (protocols, results, assays, microorganism, and so on). The same considerations are for the cytotoxicity assays with HeLa cells. Why the authors chosen a cervical cancer cell line and none a normal line? Are these experiments performed in darkness or in light irradiation conditions? In my opinion there is a contradiction between the demonstrated catalytic activity, stable in different chemical environments, of the synthetized R-CDs on the ROS production and the biological results. Please, explain why R-CDs that under light irradiation produces ROS may be not harmful for eucaryotic cells and the ambiguous “non-target organisms in aquatic environment”.
· Line 233 – How was measured the daily light absorption?
· Line 239-242 – In this sentence I recognized an apparent contradiction, I suggest rephrasing it.
· Lines 249-250 – Authors wrote that “the enhanced antialgal effect could be attributed to the abundant ions present in seawater which effectively facilitated the catalytic activity of the R-CDs” but the previous experiments performed at different temperature, pH, NaCl concentrations, metallic cations showed substantial unaltered ROS production from the control. Is it, so on, the correct interpretation?
· Experiments included in graph 4b and 4c are not discussed (protocols, experimental setup, and so on).
· Sample preparation for SEM, CLSM and FLIM imaging are missing as for the experimental parameters.
Minor editing required.
Author Response
Response to the Referees’ Comments
We thank the referees’ positive and valuable comments. The following are point-by-point to the referees’ comments/questions.
Reviewer #2:
Despite the article “Visible-Light-Activated Carbon Dot Nanoenzymes for ROS-Mediated Inhibition of Algae Growth” reveals interesting research ideas, and massive number of experiments both in the physical and chemical characterization and on the biological tests, I perceived unacceptable poor attention on the manuscript preparation. All the text is deficient of detailed descriptions of the experiments and lacks literature references supporting. It sounds very approximate on the discussion of the experimental results and on the description of graphs and images. I would like to consider the best scenario suggesting an extensive revision of the text after the publication on this journal.
Some comments are below reported:
- In the great part of the text, starting with the title, the enzymatic or nanoenzymatic activity of R-CDs is suggested. I consider inappropriate the use of this definition because it is universally attributed to biological entities – such as proteins or nucleic acids – catalysing biochemical reactions. I suggest the use of photocatalytic or catalytic activity terms to define the chemical behaviour of R-CDs when producing ROS under light stimulation.
Response 1:
Thanks for your careful and valuable comments. We have deeply considered your opinion and revised the definition mentioned in new submission.
- Sect. 2.2 – Centrifugation speed, time and instrument type are not depicted (line 91). On the end of the paragraph (lines 96-97), it is written that a “4 mg/mL dispersion in deionized water of R-CDs is used for subsequent experiments”, but it is not specified which type. Note that biological experiments were performed on completely different concentrations.
Response 2:
We are very grateful for your careful and valuable comments. The description of the centrifugal part has been supplemented in the main text as follows:
“[…] After cooling to room temperature, the solution was centrifuged (10000 rpm, 5min) and filtered to obtain clear orange solution A. Second, solution A (3 mL) and benzene sulfonamide (3 mM) were added to a mixture of deionized water (25 mL) and ethanol (2 mL), followed by heating at 180 °C for 8 h. […]”
Furthermore, we sincerely apologize for any misunderstanding caused by the description of R-CDs concentration. In reality, we merely prepared the substance at a convenient concentration for computational purposes, while subsequent experiments included comprehensive and precise concentration details. As this introductory statement might lead to misunderstanding, we have opted to remove this passage. Now the sentence comes to:
“[…] The concentration of R-CDs was determined to be 6.25 mg/mL.”
- Line 105 – Please, make sure that abbreviations have been correctly deciphered (TBN).
Response 3:
We are very grateful for your careful review. We sincerely apologize for the oversight that should have been corrected prior to submission. Once again, we truly appreciate your meticulous corrections. We have supplemented its full name and rectified the erroneous section. The revised portion is provided below:
“[...] Analytical grade 2,4-difluorobenzoic acid, lysine, benzene sulfonamide, and 3, 3’, 5, 5’-tetramethylbenzidine (TMB) as well as ethanol and metal salts (MnCl2¬, CaCl2, CuSO4, NaCl, MgCl2ž6H2O, LiCl, ZnCl2, BaCl2) were purchased from the Macklin Biochemical Co. Ltd. (Shanghai, China). [...]”
“The color reaction between TMB and ROS reflects the ROS productivity of R-CDs.”
- Line 106 – It is reported that “the intensity of the infrared absorption spectrum (OD 652) was used to characterize the ROS content”. TBN assay for ROS content quantification is colorimetric, exploiting the UV/Vis light absorption.
Response 4:
We do appreciate your positive and valuable comments. We have added experimental details regarding the measurement of ROS content.
2.4.1 Determination of the ROS content
“[...] A 1 ml detection solution will be prepared using PBS buffer to mix R-CDs and TMB, with a concentration of 0.4 mg/ml for R-CDs and 4 mM for TMB. The absorbance will be measured using UV-Vis spectroscopy after exposure to light. ”
Reference:
- Q. Zhu, M. L. Zhang, L. Pu, P. P. Gai, F. Li, Nitrogen-Enriched Conjugated Polymer Enabled Metal-Free Carbon Nanozymes with Efficient Oxidase-Like Activity. Small 18, 3(2022).
- Lines 112-113 – Please, add the standard deviation value for the average size of R-CDs.
Response 5:
We do appreciate your careful review. We have added it in the new submission.
The particle size ranged from 1.75 to 5.25 nm with an average size of 3.37 ±0.833 nm. To investigate the structural composition of the R-CDs, FT-IR analysis was conducted to determine the types of functional groups present on their surfaces.
- When attributing infrared absorption and XPS peaks some relevant references must be cited. Moreover, to the unambiguously FT-IR peaks attributions it is necessary to show the complete spectrum that is truncated around 1550 cm-1 in Fig. 1c. Looking at XPS O 1s high resolution spectrum O=C-O and C=O contributions are reported, but it seems that no typical absorptions of this functional groups are present on the FT-IR spectrum. How the authors explain this apparent contraddiction?
Response 6:
Thanks for you careful and valuable review. We have added some references in the description of FT-IR and XPS sections. Due to the substantial background noise during the FT-IR testing, the deduction of background noise resulted in the removal of several potential characteristic details. The actual existing functional groups are determined with higher precision through the XPS results.
Reference
[51] D. Kumar, G. Kumar, V. Agrawal, Green synthesis of silver nanoparticles using Holarrhena antidysenterica (L.) Wall.bark extract and their larvicidal activity against dengue and filariasis vectors. Parasitology Research 117, 2(2018) 377-389.
[52] J. Liu, J. F. Lu, J. Kan, Y. Q. Tang, C. H. Jin, Preparation, characterization and antioxidant activity of phenolic acids grafted carboxymethyl chitosan. International Journal of Biological Macromolecules 62, (2013) 85-93.
[53] F. Guo, M. Y. Li, H. J. Ren, X. L. Huang, K. K. Shu, W. L. Shi, C. Y. Lu, Facile bottom-up preparation of Cl-doped porous g-C3N4 nanosheets for enhanced photocatalytic degradation of tetracycline under visible light. Separation and Purification Technology 228, (2019).
[54] H. Li, S. Ye, J. Q. Guo, H. B. Wang, W. Yan, J. Song, J. L. Qu, Biocompatible carbon dots with low-saturation-intensity and high-photobleaching-resistance for STED nanoscopy imaging of the nucleolus and tunneling nanotubes in living cells. Nano Research 12, 12(2019) 3075-3084.
[55] T. Dufour, J. Minnebo, S. Abou Rich, E. C. Neyts, A. Bogaerts, F. Reniers, Understanding polyethylene surface functionalization by an atmospheric He/O-2 plasma through combined experiments and simulations. Journal of Physics D-Applied Physics 47, 22(2014).
[56] J. F. Weaver, G. B. Hoflund, SURFACE CHARACTERIZATION STUDY OF THE THERMAL-DECOMPOSITION OF AGO. Journal of Physical Chemistry 98, 34(1994) 8519-8524.
[57] Y. T. Zeng, Z. B. Xu, J. Q. Guo, X. T. Yu, P. F. Zhao, J. Song, J. L. Qu, Y. Chen, H. Li, Bifunctional Nitrogen and Fluorine Co-Doped Carbon Dots for Selective Detection of Copper and Sulfide Ions in Real Water Samples. Molecules 27, 16(2022).
- Line 142 – Addspectrofluorometernext to “spectrophotometer”; Line 143 – Add UV/Vis after “absorption spectrum”; Lines 144-145 – Add a reference for the 263 nm peak attribution.
Response 7:
Thank you very much for your careful inspection. We have completed the corresponding description. In addition, we checked the UV-Vis absorption peak at 263 nm and found that is should not be the π-π* transition of the C-C aromatic bond in the benzene ring, but correspond to the n-π* transition of the C=O bond. For this reason, we modified the passage in the new submission. Thanks for your careful review again.
Reference: Y. C. Zhang, C. J. Li, L. B. Sun, J. Zhang, X. J. Yang, H. Y. Ma, Defects coordination triggers red-shifted photoluminescence in carbon dots and their application in ratiometric Cr(VI) sensing. Microchemical Journal 169, (2021).
- An emission band centred around 504 nm is reported for R-CDs, but the excitation wavelength is missing and more importantly, the authors found that the excitation spectrum is centred around 462 nm but, again, the emission wavelength is missing. In addition, how the authors explains that the excitation spectrum does not match with the UV/Vis absorption spectrum?
Response 8:
Thanks for your careful review. For the sake of visual presentation, certain cropping was applied to its peripheral regions, and the complete spectrum can be observed in the Excitation map (Fig. 2b). On the other hand, it should be noted that the location of the excitation peak does indeed correspond to absorption. The magnified image of a specific region within the absorption spectrum is presented below.
- Lines 153-156 – Please specify light source type and its intensity used for R-CDs irradiation and R-CDs concentration. Excitation and emission wavelengths must be depicted. The same considerations must be extended to pH stability.
Response 9:
Thanks for your careful and valuable comments. We have provided additional information in the previous experimental section. To ensure its general applicability in the visible light range, we used a three-color mixed LED light source for illumination instead of lasers.
2.4.2 Stability experimenta
Experimental investigations were conducted to assess the photocatalytic stability of R-CDs (0.4 mg/ml) under various conditions, encompassing different temperatures (20-80 °C), buffered solutions with varying pH values (pH = 3-11), different NaCl concentrations (0-50 mg/ml), and solutions containing different ions (25 μM of Mn2+, Ca2+, Na+, Mg2+, Li+, Zn2+, and Ba2+). The R-CDs/TMB mixture dissolved in ultrapure water at 20°C served as the control group. Following 10 minutes of visible light irradiation (LED, 5000 lux), OD652 was measured to establish A0. The experimental group's OD652, denoted as A, was measured after controlling variables. The ratio A/A0 was computed, and the entire process was repeated three times for accuracy.
- Please, report the irradiation light source (type, intensity, experimental setup inside EPR spectrometer) employed for EPR experiments. Reinforce the results with literature references.
- There is no trace on the discussion about the experiments performed by EPR with DMPO trapping agent. Revise it.
Response 10,11:
Thanks for you careful review. We have added corresponding discussions in the new submission:
“[...] On the contrary, when TEMPO as the spin trapping agent and illuminate by visible light (LED, 200mW), the characteristic triplet signal (1:1:1) attributed to 1O2 can be clearly seen. Moreover, using DMPO as the spin trapping agent can also confirming the existence of žO2-[59, 60]. [...]”
Reference
[59] J. Elistratova, A. Mukhametshina, K. Kholin, I. Nizameev, M. Mikhailov, M. Sokolov, R. Khairullin, R. Miftakhova, G. Shammas, M. Kadirov, K. Petrov, A. Rizvanov, A. Mustafina, Interfacial uploading of luminescent hexamolybdenum cluster units onto amino-decorated silica nanoparticles as new design of nanomaterial for cellular imaging and photodynamic therapy. Journal of Colloid and Interface Science 538, (2019) 387-396.
[60] W. W. He, Y. T. Liu, W. G. Wamer, J. J. Yin, Electron spin resonance spectroscopy for the study of nanomaterial-mediated generation of reactive oxygen species. Journal of Food and Drug Analysis 22, 1(2014) 49-63.
- It is not clear which type of assay was utilized for the ROS production experiments in Fig. 3e, 3f, 3g, 3h, 3i. Moreover, neither the concentration of the various metal cations none the ionic strength of the solution is reported. In line 206 it is mentioned “plasma”, what the authors intended to say? In addition, who are the “non-target organisms in aquatic environment” (lake and marine) for the experiments in Fig. 3i? I can’t see any trace of discussion about the latter experiment (protocols, results, assays, microorganism, and so on). The same considerations are for the cytotoxicity assays with HeLa cells. Why the authors chosen a cervical cancer cell line and none a normal line? Are these experiments performed in darkness or in light irradiation conditions? In my opinion there is a contradiction between the demonstrated catalytic activity, stable in different chemical environments, of the synthetized R-CDs on the ROS production and the biological results. Please, explain why R-CDs that under light irradiation produces ROS may be not harmful for eucaryotic cells and the ambiguous “non-target organisms in aquatic environment”.
Response 12:
Thanks for your careful and valuable review. We supplemented these contents, about Fig. 3e, 3f, 3g, 3h, 3i, in the stability experimental section (2.4.2).
We have corrected the “plasma” to “R-CDs/TMB solution” in the new submission. Thank you once again for your meticulous and considerate review. We have carefully taken into account your concerns regarding the potential harm of R-CDs. Due to the lack of investigation into aquatic microorganisms and in the interest of experimental rigor, we have decided to remove the speculation about the harmlessness of non-target organisms in aquatic environments from the article. Regarding the use of HeLa cells for cytotoxicity assessment, this choice was based on the laboratory's practical conditions and the collective experience from previous cytotoxicity experiments. Considering that R-CDs would not receive light irradiation upon entering the human body, cytotoxicity tests were conducted in a dark environment. Below is the modified description of the Fig. 3i section:
“[...]Considering the potential use of R-CDs in natural aquatic environments, where they might enter the human body through the food chain, we conducted cytotoxicity tests using HeLa cancer cells as representatives. Since R-CDs cannot receive visible light inside the human body, the experiments were conducted in a dark environment. Fig. 3i shows the experimental results, indicating that compared to the control group, cell viability remains above 80% even at the highest working concentration. These findings suggest that R-CDs can stably generate ROS in marine environments without significant harm to public health. However, further research is needed to investigate the long-term environmental impact and potential toxicity of R-CDs.”
- Line 233 – How was measured the daily light absorption?
Response 13:
Thanks for your valuable review. It appears that our previous description may have led to misunderstandings. In fact, what we meant was to assess the density of algae by measuring their daily UV-vis absorption at 650 nm. We have made revisions in the new submission:
“[...] PGS was cultured in a conical flask under light (LED, 5009 lux), and its growth curve was plotted based on daily UV-vis absorption at 650nm (brown line in Fig. 4a). [...]”
- Line 239-242 – In this sentence I recognized an apparent contradiction, I suggest rephrasing it.
Response 14:
Thanks for your valuable comments. We modified this statement in response to 12.
- Lines 249-250 – Authors wrote that “the enhanced antialgal effect could be attributed to the abundant ions present in seawater which effectively facilitated the catalytic activity of the R-CDs” but the previous experiments performed at different temperature, pH, NaCl concentrations, metallic cations showed substantial unaltered ROS production from the control. Is it, so on, the correct interpretation?
Response 15:
We are very grateful for your valuable and careful comments. From Fig. 3h, it can be observed that Mg2+ ions slightly enhance the ROS generation capability of R-CDs. Considering the presence of numerous Mg2+ ions in marine environments, we have made a speculative inference. Taking your suggestions into full consideration and aiming to ensure the rigor of the article, we have decided to remove this conjecture.
- Experiments included in graph 4b and 4c are not discussed (protocols, experimental setup, and so on).
Response 16:
Thanks for your careful review. We supplemented its content in the experimental section (2.5 Anti-algae experiment).
17.Sample preparation for SEM, CLSM and FLIM imaging are missing as for the experimental parameters.
Response 17:
Thanks for your careful review. We supplemented its content in the experimental section (2.6 imaging experience).
Round 2
Reviewer 1 Report
It is my pleasure to recommend manuscript for publication.